# Dynamic Scale for Transformer

**Xiaoya Tang**[1]                                                 XIAOYA.TANG@UTAH.EDU
**Xiwen Li**[1]                                                       XIWEN.LI@UTAH.EDU
**Tolga Tasdizen**[1,2]                                            TOLGA@SCI.UTAH.EDU
[1] *Scientific Computing and Imaging Institute, University of Utah, SLC, UT, USA*
[2] *Electrical and Computer Engineering, University of Utah, SLC, UT, USA*

**Editors:** Accepted for publication at MIDL 2025

## Abstract

To enhance the hierarchical transformer with fixed embedding sizes, we propose a dynamic CLS token that aggregates information from CLS tokens across all layers, each embedded with varying receptive fields, by leveraging a squeeze-and-excitation module. This architecture offers a more flexible approach to utilizing multi-scale features in transformers. Code is available at: https://github.com/xiaoyatang/DynamicCLS.git.

**Keywords:** Multi-scale, Dynamic CLS token, ECG Classification

## 1. Introduction

Vision Transformers (ViT) (Dosovitskiy, 2020) are challenged by their reliance on fixed-resolution patch tokenization. Recent studies have explored hierarchical or pyramid structures within vision transformers, utilizing convolutional or pooling layers to downsample input features (Li et al., 2022). However, these approaches still rely on fixed resolutions within each stage, which restricts their representational capabilities. To overcome these limitations, novel local and window attention mechanisms have been developed to constrain the receptive field of attention, reducing the complexity of self-attention from quadratic to linear (Hassani et al., 2023; Liu et al., 2021; Dong et al., 2022). Some methods incorporate spatial-wise (Tang et al., 2024b) or channel-wise attentions (Wang et al., 2022), leveraging multi-scale representations from a CNN backbone. Others introduce spatial priors into self-attention through decay matrices based on Manhattan distances (Fan et al., 2024), or combine local attention with deformable convolutions for adaptive receptive fields (Pan et al., 2023). Despite these advances, most existing models depend on intricate attention designs or heavy CNN backbones, increasing computational complexity. In our previous work (Tang et al., 2024a), we demonstrated that a simple four-layer depth-wise CNN, trained end-to-end with a transformer, can effectively extract pyramid feature representations from ECG signals while reducing spatial resolution and maintaining channel dimensions. By progressively decreasing the size of input features at each transformer stage, we compelled the attention's receptive field to transition from local to global, leveraging the multi-scale inductive bias necessary for downstream tasks. Nevertheless, this model, like most existing ones, uses fixed feature sizes in attention mechanisms, not fully exploiting multiple scales.

To overcome this limitation, we introduce the Dynamic Scale Transformer. This innovative model dynamically aggregates CLS tokens from all layers, basing the aggregation on attention weights derived from a globally pooled representation of the input. Inspired by dynamic convolution (Chen et al., 2020), this approach allows us to achieve deformable scales

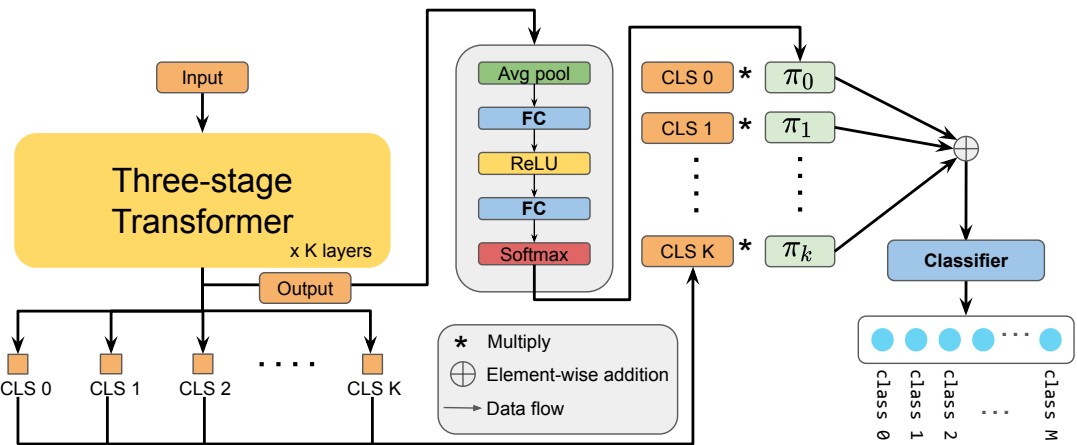

Figure 1: Dynamic scale based on a three-stage transformer.

without increasing the depth or width of the network, adding only minimal computational overhead. Our method sidesteps the need to choose between scales, instead dynamically adapting to the most informative features to enhance predictions. This aligns with the increasing trend towards dynamic and localized network designs, as dynamic weights enhance the network's inductive bias (Han et al., 2021).

## 2. Methodology

Building upon a three-stage hierarchical transformer for ECG diagnosis, we've innovated the prediction head by transitioning from a singular final-stage CLS token to a dynamic CLS token, utilizing a squeeze-and-excitation module composed of two linear layers to determine weight scores for a sequence of CLS tokens. For a given output tensor $X \in \mathbb{R}^{N \times d}$ from the transformer, along with $K$ CLS tokens from every attention layer, the process is defined as follows. Initially, average pooling condenses the global spatial information of $X$. Subsequently, two fully connected layers—with a ReLU activation in between—generate attention weights for $K$ CLS tokens, each representing an aggregated view from a specific receptive field. The scores are scaled by a temperature $\tau$ before softmax. This reflects findings by (Raghu et al., 2021), where ViT's effective receptive fields evolve from local to global across layers. With insufficient training data, lower attention layers fail to attend locally and instead focus entirely globally. Hierarchical transformers like our three-stage model partially address this by leveraging progressively shifting receptive fields and multi-scale embeddings. By computing a dynamic CLS token, which encapsulates information across different receptive fields through layers, our model adapts more effectively to different scales of features, as illustrated in Figure 1 and Eq. 1.

$$\pi_k = \text{Softmax}\left(\text{FC}(\text{ReLU}(\text{FC}(\text{AvgPool}(X))))/\tau\right), \tau > 0$$

$$\text{s.t.} \quad 0 \leq \pi_k(x) \leq 1, \ \sum_{k=0}^{K} \pi_k(x) = 1, \ \text{Dynamic\_CLS} = \sum_{k=0}^{K} (\pi_k \cdot \text{CLS}_k) \tag{1}$$

## 3. Results and Analysis

We evaluate our model on the 2020 PhysioNet/CinC Challenge dataset (Alday et al., 2020), utilizing a 10-fold validation for the multi-label classification of 24 diagnoses. We reported macro $F_\beta$ and $G_\beta$ - weighted harmonic means of precision and recall — and the challenge score S, which credits correct diagnoses and penalizes incorrect ones based on the similarity between arrhythmia types. Experimental details can be found in Appendix A. According to Table 1, we conjecture that our model may be improving in identifying positive cases (true positives) while reducing false alarms (false positives), as indicated by the increase of $G_\beta$. The decrease in $F_\beta$ might suggest that the model is reducing false positives at the cost of increasing false negatives. This could indicate the model is more conservative in its predictions. The improvement in challenge metric shows the model with dynamic CLS token is correctly classifying more instances, distinguishing between different classes more accurately. Additionally, we experimented with two different temperatures and found that they influence model performance by adjusting the sharpness of attention distributions.

| Model | Fbeta | Gbeta | Challenge | Size(M) |
|---|---|---|---|---|
| LSTM | 0.4323 ($\pm$0.55%) | 0.2742 ($\pm$1.90%) | 0.4372 ($\pm$1.67%) | - |
| CNN | 0.4519 ($\pm$1.55%) | 0.2862 ($\pm$2.90%) | 0.4542 ($\pm$1.68%) | - |
| ResNet | 0.5088 ($\pm$0.41%) | 0.3278 ($\pm$2.69%) | 0.5158 ($\pm$0.80%) | - |
| ViT | 0.3263 ($\pm$1.65%) | 0.1970 ($\pm$1.88%) | 0.3197 ($\pm$2.44%) | - |
| Swin Transformer | 0.4812 ($\pm$0.87%) | 0.3045 ($\pm$0.66%) | 0.4811 ($\pm$1.41%) | - |
| BaT (Li et al., 2021) | 0.5011 ($\pm$0.68%) | 0.3125 ($\pm$1.15%) | 0.4958 ($\pm$0.83%) | - |
| Res-SE (Zhao et al., 2020) | 0.5607 ($\pm$1.30%) | 0.3264 ($\pm$2.94%) | 0.5939 ($\pm$0.30%) | 8.84 |
| SpatialTemporalNet | 0.4296 ($\pm$2.82%) | 0.2403 ($\pm$2.98%) | 0.4322 ($\pm$9.81%) | 4.52 |
| Prna (Natarajan et al., 2020) | 0.4975 ($\pm$5.17%) | 0.2679 ($\pm$6.99%) | 0.5463 ($\pm$3.23%) | 13.64 |
| Prna + CLS_Token | 0.5211 ($\pm$1.38%) | 0.2926 ($\pm$2.46%) | 0.5732 ($\pm$2.11%) | 13.64 |
| Transformer (Tang et al., 2024a) | **0.5850** ($\pm$**1.15**%) | 0.3459 ($\pm$0.38%) | 0.6063 ($\pm$0.18%) | 18.63 |
| Our($\tau = 7$) | 0.5811 ($\pm$1.93%) | **0.3501** ($\pm$**2.54%**) | 0.6069 ($\pm$0.65%) | 18.65 |
| Our($\tau = 34$) | 0.5798 ($\pm$0.34%) | 0.3420 ($\pm$1.01%) | **0.6224** ($\pm$**0.10%**) | 18.65 |

Table 1: Performance on multi-label ECG diagnosis. Results for all models were averaged over three folds in 10-fold cross-validation unless otherwise indicated by the source. The best performance is highlighted in bold, and the second best is underlined.

## 4. Conclusion

In this work, we introduce a dynamic learned attention applied to CLS tokens from all layers of a transformer, fully leveraging the progressively shifting receptive fields inherent in these tokens. We evaluate our module on a multi-label ECG diagnosis task, demonstrating the effectiveness of the dynamic CLS token mechanism with trivial computational overhead. This approach can be seamlessly integrated into various transformer-based models, offering a novel method for dynamically utilizing multi-scale information.

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

## Appendix A. Data and Evaluation Metrics

We utilize the public training data from the 2020 PhysioNet/CinC Challenge (Alday et al., 2020). The public dataset comprises $43,101$ recordings, and we adopt the 10-fold split used by the winner model 'Prna' (Natarajan et al., 2020). This setup involves a multi-label classification task related to 24 diagnoses. Following the preprocessing steps of 'Prna', we resample all recordings to $500Hz$, apply an FIR bandpass filter, and perform normalization. We also randomly crop multiple fixed-length ECG segments of $T = 15$ seconds from the input, adding padding when necessary for segments shorter than 15s. We also leveraged the wide features that they used. For evaluation metrics, we report macro $F_\beta$, $G_\beta$, geometric mean(GM) combining precision and recall and the challenge score defined by the challenge organizers (Alday et al., 2020), detailed in Eq. 2. The score $S$ generalizes standard accuracy by fully crediting correct diagnoses and penalizing incorrect ones based on the similarity between arrhythmia types. Here $a_{ij}$ represents an entry in the confusion matrix corresponding to the number of recordings classified as class $c_i$ but actually belonging to class $c_j$, with different weights $w_{ij}$ assigned based on the similarity of classes $c_i, c_j$:

$$F_\beta = (1 + \beta^2) \cdot \frac{TP}{(1 + \beta^2) \cdot TP + FP + \beta^2 FN}$$
$$G_\beta = \frac{TP}{TP + FP + \beta FN}, \beta = 2 \tag{2}$$
$$S = \sum_{ij} w_{ij} a_{ij}$$

### A.1. Implementation and Training Details

We implemented our model using `PyTorch` and trained it on four NVIDIA TITAN RTX GPUs, each with 12 GB of memory. The training batch size was set to 96, and the evaluation batch size to 256. Model parameters were optimized using the Adam optimizer. Our training configuration, including the learning rate and learning rate scheduling strategy, follows the setup described in (Natarajan et al., 2020). We monitored the training process using the multi-label cross-entropy loss function.

