# OpenReview forum: "Dynamic Scale for Transformer"
_MIDL.io/2025/Short_Papers — MIDL 2025 - Short Papers_

### Official Review · Reviewer_pjiF · 2025-04-28

**Rating:** 4
**Confidence:** 4

**Summary:**

This paper presents a dynamic scale transformer to fully utilize the progressively shifting receptive fields inherent in CLS tokens. The proposed dynamic CLS token aggregates information across different receptive fields through all layers. The proposed method has been demonstrated on a multi-label ECG diagnosis task, and shows higher performance than the baseline methods.

**Strengths:**

- The proposed method can exploit multi-scale features, which addresses a limitation of the existing transformer methods in that they use fixed feature sizes in attention mechanisms.
- The experimental results using the 2020 PhysioNet/CinC Challenge dataset show that the proposed method provides fewer false positives and higher performance on 24-class classification.

**Weaknesses:**

- The performance improvement of the proposed method is marginal compared to the second-best method.
- As the proposed method uses information of the multi-scale receptive fields, it requires higher complexity than the existing transformer-based methods.

---

### Decision · Program_Chairs · 2025-05-01

Accept